# Thymol Edible Coating Controls Postharvest Anthracnose by Regulating the Synthesis Pathway of Okra Lignin

**DOI:** 10.3390/foods12020395

**Published:** 2023-01-13

**Authors:** Qinqiu Zhang, Zhuwei Wang, Yinglu Li, Xinzhi Liu, Lang Liu, Jing Yan, Xinjie Hu, Wen Qin

**Affiliations:** College of Food Science, Sichuan Agricultural University, Ya’an 625014, China

**Keywords:** postharvest okra anthracnose, *C. fioriniae*, thymol edible coating, lignin synthesis pathway

## Abstract

Okra has received extensive attention due to its high nutritional value and remarkable functional characteristics, but postharvest diseases have severely limited its application. It is important to further explore the methods and potential methods to control the postharvest diseases of okra. In this study, *Colletotrichum fioriniae* is the major pathogen that causes okra anthracnose, which can be isolated from naturally decaying okra. The pathogenicity of *C. fioriniae* against okra was preliminarily verified, and the related biological characteristics were explored. At the same time, an observational study was conducted to investigate the in vitro antifungal effect of thymol edible coating (TKL) on *C. fioriniae*. After culturing at 28 °C for 5 days, it was found that TKL showed an obvious growth inhibition effect on *C. fioriniae*. The concentration for 50% of the maximal effect was 95.10 mg/L, and the minimum inhibitory concentration was 1000 mg/L. In addition, it was found that thymol edible coating with a thymol concentration of 100 mg/L (TKL100) may cause different degrees of damage to the cell membrane, cell wall, and metabolism of *C. fioriniae*, thereby inhibiting the growth of hyphae and causing hyphal rupture. Refer to the results of the in vitro bacteriostatic experiment. Furthermore, the okra was sprayed with TKL100. It was found that the TKL100 coating could significantly inhibit the infection of *C. fioriniae* to okra, reduce the rate of brown spots and fold on the okra surface, and inhibit mycelium growth. In addition, the contents of total phenols and flavonoids of okra treated with TKL100 were higher than those of the control group. Meanwhile, the activities of phenylalaninammo-nialyase, cinnamic acid-4-hydroxylase, and 4-coumarate-CoA ligase in the lignin synthesis pathway were generally increased, especially after 6 days in a 28 °C incubator. The lignin content of TKL-W was the highest, reaching 65.62 ± 0.68 mg/g, which was 2.24 times of that of CK-W. Therefore, TKL may promote the synthesis of total phenols and flavonoids in okra, then stimulate the activity of key enzymes in the lignin synthesis pathway, and finally regulate the synthesis of lignin in okra. Thus, TKL could have a certain controlling effect on okra anthracnose.

## 1. Introduction

Okra is a traditional vegetable crop originating from India. It has excellent planting characteristics and is rich in nutrient elements, such as various vitamins, minerals, polyphenols, polysaccharides, etc. [1]. It is commonly used as a functional vegetable. It is widely distributed in Asia and Africa, and its tender pod is the main edible part [2]. Thanks to the continuing rise of health and nutritional foods in recent years, okra, as a front-runner among vegetables with high nutritional value, has been favored by the food, processing, pharmaceutical, cosmetics, and other industries [1,3]. However, fresh okra fruit pods are prone to postharvest diseases and spoilage due to their properties and various external factors and lose their nutritional value and edible value [4,5]. One such disease is due to *Mucor.* sp. caused by postharvest soft rot [6]. In particular, the brown rot caused by *Aspergillus niger* (irregular brown spots on the surface, soft and rotten tissues, and even discharge of the juice) is okra’s most typical postharvest disease, bringing about great economic losses to okra and hindering its industrialization process [7]. However, the prevention and control measures for postharvest brown rot of okra are very few at present. The traditional pesticide control effect is not ideal and has serious food safety problems [8].

*Colletotrichum* is mainly distributed in tropical and temperate regions around the world and comprises more than 189 species, which is a huge worldwide Colletotrichum family [9]. Colletotrichum species can cause anthracnose and fruit rots on extensive fruit crops, especially banana, strawberry, citrus, avocado, papaya, mango, and apple [10]. *C. fioriniae* of the C. acutatum species complex is an important hemibiotrophic pathogen of vegetables and fruits in temperate regions worldwide [11]. It can harm the roots, stems, leaves, flowers, fruits, and other parts of the plant. *C. fioriniae* is a member of the genus Anthracis. According to the existing research findings, *C. fioriniae* causes postharvest bitter rot in apples in south Tyrol (Northern Italy) [12]; infects invasive Japanese hop (*Humulus scandens*) in the United States [13]; and causes anthracnose in pecan [14]. Therefore, it is essential to prevent and control the anthrax disease of fruits and vegetables caused by *C. fioriniae*.

Furthermore, okra undergoes a series of physiological and biochemical reactions during transportation and storage due to the postharvest ripening process, which leads to lower quality and economic loss [15]. Moreover, lignin, a compound synthesized by the metabolism of phenolic substances, commonly exists in all kinds of fruits and vegetables, mainly in the secondary wall of fruit and vegetable cell walls [16]. It gradually accumulates with the metabolism of phenolic substances, thus increasing the peel thickness and affecting the organoleptic quality and time–temperature tolerance of fruits and vegetables [17]. Immense amounts of concrete studies have demonstrated that polyphenols are one of the most important physiological active substances in plant tissues and have various physiological functions such as antioxidant, anti-obesity, hypoglycemic, and anti-gastrointestinal-inflammation [18]. Phenols are the precursors of lignin synthesis, and their contents and types directly affect the lignification process of fruits and vegetables. Meanwhile, preliminary studies have proved that okra is rich in physiologically active substances such as polyphenols and polysaccharides, which accumulate with the growth and maturation of fruits [19,20], and show a variety of physiological functions including antioxidant, anti-obesity, hypoglycemic, and anti-gastrointestinal-inflammation. However, phenolic substances are easily degraded and transformed, resulting in increased lignification of okra and rapid deterioration of its sensory and nutritional quality [21]. How to reduce the degradation and transformation of phenols is also the key issue of okra storage after harvest. Regulating the metabolism of phenols in the lignification pathway is an important method to solve this problem.

Okra lignin can be divided into three types: S, G, and H. The metabolic pathway first forms aromatic amino acids through the shikimic acid pathway; then, the aromatic amino acids undergo deamination, carboxylation, and methylation [22]. A series of reactions such as methylation were used to synthesize hydroxycinnamate compounds and hydroxycinnamate acyl CoA compounds; finally, hydroxycinnamate compounds and hydroxycinnamate acyl CoA compounds were reduced to various lignin in the phenylalanine pathway. Alcohol monomers and lignol monomers are further polymerized to form lignin [23]. Therefore, regulating the activities of enzymes involved in the phenylalanine pathway can significantly affect the accumulation of lignin in fruits and vegetables [24,25]. Phenylalanine ammonia-lyase (PAL) is a key prerequisite enzyme for the phenolic synthesis of okra lignin. Moreover, previous studies have shown that the accumulation of lignin in okra postharvest pods is mainly the accumulation of cinnamic acid [26], and the synthesis of lignin from cinnamic acid is achieved mainly through two enzyme action sites, namely cinnamic acid-4-hydroxylase (C4H) and 4-coumarate-CoA ligase (4CL).

Since the 21st century, plant extracts have been widely used to prevent and preserve fruit and vegetable diseases [27]. These plant extracts, such as cinnamic aldehyde [28,29] and carvacrol [30], thymol, menthol [31], and vanillin [32], all have a good antifungal effect. These extracts have been widely applied to the preservation and corrosion protection of citrus fruit, berries, nuts, and all kinds of leaf vegetables; they have achieved good results and some plant extracts, antibacterial card paper, and crisper have also begun to circulate on the market [33]. In particular, thymol has been used in various forms in the prevention and control of fruit and vegetable diseases and storage in recent years. It is mainly used in the form of edible coating for fruit and vegetable preservation and disease prevention and control [34,35,36]. For example, Hp-β-cyclodextrin-encapsulated thymol microcapsules can be used as a substitute for synthetic fungicides to induce lemon resistance to sour rot decay; it has the effect of preventing and controlling fusculosis of citrus fruit [37]. The coating, which is composed of chitosan and thymol, can maintain the quality of fresh fig fruit, reduce the rate of fungal impregnation and decay, and extend the shelf life of fruit [38]. Blending thymol into a commercial shellac coating and applying it to “ruby red” grapefruit inoculated with *L. theobromae* was found to significantly inhibit stem rot and extend the shelf life of the grapefruit [39]. Thymol/sodium alginate composite film can effectively reduce the weight of fresh-cut apple slices, retain nutrients and surface color, and inhibit the growth and reproduction of *Staphylococcus aureus* and *Escherichia coli* [40]. Therefore, it can be concluded that the composite coating material of thymol has been widely used in the preservation of various fruits and vegetables and the prevention and control of postharvest diseases. The material provides good preservation, disease resistance, and inhibition of postharvest pathogens [41] for all kinds of fruits and vegetables.

In this study, thymol, emulsifier, beta-cyclodextrin, and polyethylene glycol (PEG6000) as the main raw material were used and mixed in proportion with a spray powder preparation of thyme phenol microcapsules. Then, the substrate liquid of konjac glucomannan (KGM) and low acyl gellan gum (LG) was added in, and the thymol edible coating was formed. This procedure was followed to further explore the mechanism of how thymol edible coating (TKL) controls postharvest anthracnose by regulating the lignin synthesis pathway of okra. It is expected to provide new technologies and solutions for the prevention and control of postharvest okra diseases and the extension of its freshness period.

## 2. Materials and Methods

### 2.1. Materials and Chemicals

The materials used in the present study were as follows: KGM (Henan Wanbang Industrial Co., Ltd., Shangqiu, China); LG (Dancheng Caixin Sugar Co., Ltd., Dancheng, China); calcium stearyl lactylate (Henan Hainasen Food Technology Co., Ltd., Hainasen, China); thymol (Shanghai Yien Chemical Technology Co., Ltd., Shanghai, China); macrogol 6000 (Chengdu Colon Chemical Co., Ltd., Chengdu, China); beta-cyclodextrin (MengzHou Huaxing Biochemical Co., Ltd., Mengzhou, China); and sucrose fatty acid esters (Liuzhou Aegefu Food Technology Co., Ltd., Liuzhou, China). Fresh okra was sourced from Qingbaijiang Plantation in Chengdu, Sichuan Province, China. Naturally decaying okra was provided to Sichuan Agricultural University for pathogenic bacteria isolation. Potato dextrose agar medium (PDA) (Beijing Aoboxing Biotechnology Co., Ltd., Beijing, China) was used as a fungal culture medium. Thymol (purity 98.00%) was purchased from Shanghai Yien Chemical Technology Co., Ltd., Shanghai, China. All other chemicals and reagents are analytical grade and purchased from Chengdu Cologne Chemical Reagent Company (Chengdu, China).

Thymol microcapsules based on KGM and LG were prepared by the following process. First, a 5 g sucrose fatty acid ester was dissolved in 125 mL distilled water at 90 °C to obtain homogeneous solution A. Then, 30.79 g polyethylene glycol 6000 was added to 125 mL distilled water at 100 °C, homogenized at high speed for 1 min, and then dissolved homogeneously and degassed by ultrasonication to obtain solution B. The solution was cooled to room temperature and set aside. Then, 61.57 g of β-cyclodextrin was dissolved in 250 mL distilled water at 90 °C until clear to obtain solution C. Solutions A and B were combined and stirred at 600 r/min on a magnetic stirrer for 30 min to obtain solution D. The solution was allowed to cool to room temperature and reserve. Finally, 2.64 g thymol was dissolved in 5 g anhydrous ethanol while solution D was placed on a temperature-controlled magnetic stirrer at 40 °C and 600 r/min. The thymol-anhydrous ethanol solution was added drop by drop (2 drops/s) to the central vortex of solution D, which was stirred continuously for 3 h. The capsule formation process involved spray drying, the inlet temperature of 180 °C, and the inlet speed of 800 mL/h; and the resulting microcapsules were dried at 180 °C. In the prepared microcapsules, the effective content of thymol was 26.1 mg/g. KGM and LG-based edible coatings were combined with a 0.22% thymol microcapsule to make TKL.

Preparation of membrane matrix solution was as follows. The first step was boiling 500 mL ultrapure water + 0.125 g Konjac glucomannan (KGM) + 0.25 g low acyl knot gels (LG). The solution was then processed with a high-speed agitator (1200–1600 r/min), followed by stirring for 30 min. Then, 0.2 g calcium stearate lactate was added and stirred continually for 5–10 min. The solution was then covered with plastic wrap and brought to a boil. The homogenized membrane matrix solution was obtained by ultrasound for 30 min. The homogenized membrane matrix solution (10 mL + 1.88679 g microcapsule powder) was mixed into the homogenized TKL mother solution (thymol concentration was 5 mg/mL). Then, the solution was diluted with membrane matrix to obtain TKL of different thymol concentrations. Figure 1 shows the preparation process of thymol edible coating.

### 2.2. Isolation and Identification of Pathogens

The pathogens were isolated and purified by separating plant lesion tissue [42]. Part of the lesion tissue (6 mm^2^) was cultured on a PDA medium, a single colony was isolated and purified, and the corresponding micromorphological characteristics of the colony were observed under different magnification microscopes [43]. The single colony was cultured on PDA medium at 28 °C for 3–4 days, the hyphae were scraped, and DNA was extracted by TSINGKE DNA Extraction Kit (universal). The extracted DNA samples were diluted and amplified as PCR templates and a 1 × TSE101 gold mixture. The prepared PCR products were sequenced (the sequencing primer was ITS). The work was entrusted to the Chengdu Branch of Beijing Musculus Biotechnology Co., LTD., Beijing, China. The comparison was performed in the National Center for Biotechnology Information (NCBI) database. The species with the highest similarity were selected, and the related phylogenetic trees were constructed using MEGA 5.02 software.

### 2.3. Preliminary Verification of Pathogenicity

The purified pathogens were inoculated on healthy okra by puncture inoculation to observe the incidence of okra.

### 2.4. Observation of Biological Characteristics of Pathogens

Isolated and identified pathogens were inoculated on a PDA medium for dark culture at 28 °C for 3 days, and fungus blocks with diameters of 6 mm were taken to PDA mediums. Different carbon and nitrogen sources, temperature, pH, NaCl concentration, and light conditions were set up in the experiment. After 3 days, the colony diameter was measured by the cross-crossing method. 

### 2.5. In Vitro Antifungal Activity of TKL on Mycelium Growth and Spore Germination

The growth rate method was used to determine the effect of TKL coating solution with different concentrations of thymol on the growth of pathogenic fungus. The final concentration gradients of thymol were set as 0 mg/L, 20 mg/L, 40 mg/L, 60 mg/L, 80 mg/L, 100 mg/L, and 120 mg/L. Medium blocks containing pathogenic fungus with a diameter of 6 mm were inserted into a PDA medium with different concentrations of thymol TKL coating solution using a sterile hole punch. The culture was then incubated in a fungus incubator at 28 °C for 3 days. The colony diameter was measured by the cross-crossing method, and the corresponding growth inhibition rate was calculated.

The thymol concentrations in the TKL coating solution were set as 0, 40, 80, 100, 200, 400, 800, 1000, and 2000 mg/L, respectively. Distilled water, film-forming matrix, and anhydrous ethanol were used as controls. An appropriate amount of spore liquid was added, and the concentration was 10^5^ CFU/mL. After mixing with the TKL coating solution evenly, 200 uL was taken and placed into 96-well plates, respectively. The OD600 value of each gradient treatment was detected by a fluorescent enzyme label after 72 h culture in a 28 °C incubator with constant temperature and humidity. The minimum inhibitory concentration (MIC) of TKL on the growth of the pathogenic fungus was obtained.

### 2.6. Inhibition of TKL on the Mycelium Weight of Pathogenic Fungus

Conical flasks, which had a volume of 250 mL, containing 100 mL of liquid PDA were used as the basal medium. An appropriate amount of sterile water was added to the activated species slant to make a fungal suspension or spore suspension of 10^5^–10^6^ CFU/mL. In the experimental group, a TKL coating solution was added to the culture medium, and the thymol concentration was 100 mg/L. Blank mediums were used as a control group. The cultures were incubated in a shaker at a temperature of 28 °C to 30 °C and a rotation rate of 140 r/min. The conical flasks were removed every 12 or 24 h, and the medium was aspirated and filtered with quantitative filter paper to obtain a certain quantity of fungi. The mycelium was baked at 60–80 °C to a constant weight, and the dry weight of the hyphae was determined after weighing and peeling. Corresponding growth curves were plotted according to incubation time (h) and dry weight (g).

### 2.7. The Effects of TKL on the Structure of Pathogenic Fungus Was Observed by SEM

Refer to the method of Zhang et al. to prepare fungal culture cover glass [6]. After the prepared fungal culture glasses are dried and coated at the critical point, the results are observed with SEM, and the steps are photographed for later observation. (Equipment model: ELECTRON microscope: United States FEIF50 energy spectrum: United States EDAX OCTANE SUPER. This step was operated by E Testing Company.)

### 2.8. Effects of TKL on the Intracellular Structure and Enzyme Activity of C. fioriniae

Various enzyme activities were determined with MDA, PG, PMG, β-1.3-glucanase, chitinase, and Cx kits (These kits were purchased from Jiangxi Jingmei Biotechnology Co., LTD., Jingmei, China).

### 2.9. Effects of Postharvest Coating Wounds of TKL Inoculated with C. fioriniae on Stress Resistance of Okra

First, healthy okra were taken and divided into two groups. One group is treated with TKL coating, and the other group is treated with distilled water. The okra specifications are the same, and natural drying is used. The fresh strains that were continuously cultured on a PDA medium for 2 generations were used for testing. After growth on the medium for 3–4 days (no spores were produced), a sterile hole punch was used to take a 6 mm-diameter fungal block. The wounded inoculation was taken, and a sterilized toothpick was used to prick 6 holes into the dried fruit (the holes are concentrated together) and inoculate the fruit without injury as a comparison. There were four groups: TKL coating okra was inoculated with injury (TKL-Y); TKL coating okra was inoculated without injury (TKL-W); okra was treated with distilled water and inoculated with injury (CK-Y); and okra was treated with distilled water and inoculated without injury (CK-W). The treated okra was cultured in a constant temperature incubator at 28 °C and 75%, and samples were taken every day for a total of 14 days. The white hyphae formation rate, brown spot rate, surface fold rate, and hardness of okra were observed and recorded every day. At the same time, 4 groups were sampled at 1, 2, 3, 4, 5, 6, 7, 8, 9, and 10 days after inoculation, and the okra-related stress resistance indexes were determined.

#### 2.9.1. Hardness Determination

The fruit was measured by physical property analyzer and a TPA model. The measurement conditions were as follows: probe P/5, pre-measurement speed 2 mm/s, measurement speed 1 mm/s, post-measurement speed 2 mm/s, probe depression distance 4 mm, and force unit kg/cm^2^. Five okras were measured for each treatment group, three points were measured for each okra fruit, and the average value was taken.

#### 2.9.2. Determination of Total Phenols and Total Flavonoids

The okra sample treatment was as follows: okra slices under different treatment conditions were classified and stored in the refrigerator at −70 °C. After freezing completely, the slices were freeze-dried and crushed through a 60-mesh sieve, and the freeze-dried powder of the okra was stored in the refrigerator at −20 °C. Next, 1.0 g of ground okra powder (ground in a mortar under liquid nitrogen, with the consideration of whether it is sieved or not according to the situation) was accurately weighed, and 30 mL 70% acidified methanol (0.1% HCl, *V*/*V*; 70 mL methanol solution, fixed volume with 0.1% HCl to 100 mL) solution was added. Then, the mixture was extracted by ultrasound at room temperature (50 kHz, 480 W) for 30 min each time. After centrifugation (8000× *g*, 15 min), the supernatant was combined for subsequent determination [44].

The determination of total phenolic content (TPC) was based on the method of Liu et al. [45] with a slight modification, and 250 μL of okra fruit extract or gallic acid standard solution was added to 1.25 mL of Folin phenol reagent. The mixture was then reacted for 3 min at room temperature in the dark, and 1.25 mL of NaCO_3_ (20%, *w*/*v*) was added. After 30 min of dark reaction at room temperature, the absorbance was measured at 765 nm using a Varioskan Flash full-wavelength microplate reader (Thermo Fisher, Waltham, MA, USA). Total phenolic content was expressed as milligrams of gallic acid equivalents per gram of okra fruit dry weight (mg GAE/g DW). Next was the determination of total flavonoid content. The total flavonoid content (TFC) of okra fruit extract was determined according to the method of Lin et al. [46], which involves adding 100 μL okra extract or rutin standard solution to 30 μL 5% NaNO_2_ solution (*w*/*v*). After 6 min, 30 μL of 10% Al (NO3)3 (*w*/*v*) was added. Subsequently, 400 μL of 4% NaOH (*w*/*v*) was added and reacted at room temperature for 25 min. Finally, the absorbance of the mixture was measured at 510 nm, and the total flavonoid content was expressed as milligrams of rutin equivalents per gram of okra fruit dry weight (mg RE/g DW).

#### 2.9.3. Key Enzyme Index of Okra Lignin Synthesis

For the determination of phenylalanine ammonia-lyase (PAL) activity, refer to the method of Huang et al. [47]. The fresh pulp of the upper, middle, and lower parts of the okra fruit was taken, ground, and mixed. A total of 1.0 g of pulp tissue from it was weighed, and 9 mL of extraction buffer (0.2 mol/L boric acid buffer, containing 5 mmol of mercaptoethanol, 1 mmol of EDTA, 1 mmol of PVP, pH 8.8) was added. The resulting solution was ground in an ice-bath mortar. After homogenization, the cells were centrifuged at 8000 r/min and 4 °C for 20 min, and the supernatant was collected for use. Enzyme activity determination was as follows: 2 mL of 0.2 mol/L pH 8.8 borax buffer, 1 mL of 20 mmol/L L-phenylalanine, and 1 mL of enzyme extract were reacted at 37 °C for 1 h. The absorbance at 290 nm wavelength was measured, and the enzyme solution boiled for 1 min was used as a control. II. The activity of cinnamyl alcohol-4-hydroxylase (C4H) was measured according to the method of Sykes and Xiang et al. [48]. The fresh pulp of the upper, middle, and lower parts of okra fruit was taken, ground, and mixed. In total, 1 g of pulp tissue from it was weighed, and 9 mL of extract (0.05 mol/L pH 8.9 Tris-HCl, 15 mmol/L mercaptoethanols, 4 mmol/L MgCl2, 5 mmol/L Vc, 2% PVP) was added on ice. The bath was ground to homogenate and centrifuged at 4 °C and 8000 r/min for 20 min, and the supernatant was collected as C4H crude enzyme solution. Then, 0.2 mL of crude enzyme solution was taken, and 3.6 mL of reaction solution (2 mmol/L trans-cinnamic acid, 50 mmol/L pH 8.9 Tris-HCl, 2 mmol/L NADP-Na2) was added; the resulting solution was reacted at 25 °C for 30 min. The reaction was terminated by adding 0.2 mL of 6 mol/L hydrochloric acid solution. The absorbance at the wavelength of 340 nm was measured immediately, and the buffer solution was used for the blank control group instead of the reaction solution. Under this condition, the change of absorbance per hour was 0.01 as 1 unit of enzyme activity. Next, the activity of 4-coumaroyl CoA ligase (4CL) was measured according to the method of Tao et al. [49]. The upper, middle, and lower fresh pulp of okra fruit were taken, ground, and mixed. In total, 1 g of pulp tissue from the ground homogenate was weighed and transferred to a centrifuge tube. An amount of 9 mL of extract was added, the resulting solution was mixed in an ice bath to extract for 1 h, with irregular shaking during this period. Then, centrifugation was performed at 8000 r/min for 10 min at 4 °C, and the supernatant was the crude enzyme extract for the determination of 4CL activity. The determination of enzyme activity was as follows. It consists of p-coumaric acid 5 mmol/mL, ATP 50 mmol/mL, CoA-SH 1 mmol/mL, and MgSO_4_ 7H_2_O 15 mmol/mL. In total, 0.4 mL of enzyme solution was added during the reaction, and the reaction was carried out in a water bath at 40 °C for 10 min. Absorbance was measured at a wavelength of 333 nm.

#### 2.9.4. Determination of Okra Lignin Content

The acetylation method is used to determine the acetylation of the phenolic hydroxyl groups in the lignin [50,51], which has a characteristic absorption peak at 280 nm, and the absorbance value at 280 nm is positively correlated with the lignin content. Sample preparation was as follows. An appropriate amount of okra tissue samples was dried and ground, then passed through a 40-mesh sieve for use. An amount of 1.5 mg of sifted powder tissue was taken, 1.5 mL of 80% ethanol was added, and the combination was vortexed and mixed well. The mixture was placed in a 50 °C water bath for 20 min (shaken a few times at 3 min intervals), taken out of the running water to cool, and vortexed at 12,000 rpm and 25 °C for 10 min. The supernatant was discarded, and the precipitate was left (it is important to try to keep the precipitate). Then, 1 mL of 80% ethanol was added to the precipitation and mixed for 2 min. It was then placed in a 50 °C water bath for 20 min (shaken a few times at 3 min intervals), taken out of the running water, and cooled. It was then vortexed at 12,000 rpm/min and 25 °C for 10 min. The supernatant was discarded, and the precipitate was kept (it is important try to keep the precipitate). The precipitate was then dried at 95 °C and set aside. The reaction assay was performed using a lignin kit (96-well microplate method) (purchased from Wuhan Addison Antibiotic Technology Co., Ltd., Wuhan, China).

## 3. Results

### 3.1. Isolation and Identification of Pathogens

The lesions of anthracnose okra first appear as dark brown spots on the surface of the okra, and then the surface of the okra begins to shrink. Over time, the spots gradually become dark brown depressions that expand into round dark brown patches or ovals. Lesions present with certain greyish-white fungal hyphae (Figure 2A). The pathogenic fungus colonies on the PDA medium were round and white-brown as a whole. They presented with neat edges and vigorous aerial hyphae. They were greyish-white and were spreading to the surrounding area, and they were reddish-brown when spores were produced (Figure 2B). The hyphae were slender and transparent without a septum, and the surface was smooth and transparent (Figure 2C).

If sequences in the ITS1-5.8S-ITS4 region are compared, the isolated pathogenic fungus had the highest similarity with *C. fioriniae* (Figure 2D).

### 3.2. Pathogenicity Determination and Biological Characteristics of C. fioriniae

Acupuncture was performed on okra to verify the pathogenicity of C. fioriniae. As shown in Figure 3A, 72 h after inoculation with *C. fioriniae*, there were obvious brown patches on the surface of okra and shrinkage depressions in the corresponding parts. The surrounding tissue began to brown and rot, and the internal tissue also began to present with rot symptoms. The colony growth under different carbon source conditions was as follows. The seven carbon sources shown in Figure 3B can all make *C. fioriniae* grow; the growth rate of *C. fioriniae* with mannitol, sucrose, and starch as carbon sources was the fastest (*p* < 0.05), and the colony diameter could reach 25 mm after 3 days of culture at 28 °C. However, when fructose was used as a carbon source, the colony diameter was the smallest, at only 16.13 ± 1.14 mm. Colony growth under different nitrogen source conditions: As shown in Figure 3C, the seven nitrogen sources shown in the figure can enable *C. fioriniae* to grow. When the colony is under different nitrogen sources and is cultured at 28 °C for 3 days, the colony diameters have significant differences (*p* < 0.05). Under the condition of potassium nitrate, the colony diameter of *C. fioriniae* could reach 25.68 ± 1.07 mm; the rest were sodium nitrate, trypsin, beef paste, ammonium chloride, and urea, and ammonium sulphate was the smallest. Colony growth was as follows at different pH values. In Figure 3D, *C. fioriniae* could grow well in the pH range of 4.0–10.0, but the growth conditions were significantly different (*p* < 0.05). When the pH was 7, the growth rate was the largest, and the colony diameter after culturing at 28 °C for 3 days could reach 33.20 ± 1.17 mm, which was significantly higher than that under other pH conditions (*p* < 0.05). When the pH was 4, the growth rate was the slowest at only 19.14 ± 0.97 mm. Colony growth under different NaCl (%) conditions was as follows. In Figure 3E, *C. fioriniae* can grow well in the range of 0–3.0% NaCl concentration. When the concentration was 0.5%, the growth rate was the fastest, and the colony diameter could reach 32.39 ± 1.20 mm (*p* < 0.05). However, when the NaCl concentration was higher than 2.0%, the colony diameter was significantly lower than that of the blank control group (*p* < 0.05). Colony growth at different culture temperatures was as follows. It can be seen from the figure that *C. fioriniae* grows best under the temperature condition of 28 °C, and the colony diameter was significantly higher than that of 15 °C, 20 °C, and 24 °C (*p* < 0.05), as shown in Figure 3F. Colony growth under different light conditions was as follows. In Figure 3G, it could be seen from the figure that under 24 h light conditions, the colony diameter was the largest and could reach 25.09 ± 0.81 mm. The colony diameter was significantly reduced under half-light and dark conditions (*p* < 0.05).

### 3.3. Inhibitory Effect of TKL on C. fioriniae In Vitro

The antifungal activity of TKL against *C. fioriniae* was observed, and it was found that *C. fioriniae* was sensitive to the toxicity of thymol. Compared with the control group, different concentrations of TKL inhibited the growth of *C. fioriniae*, and the higher the concentration, the more obvious the inhibitory effect. The results showed that 20–120 mg/L thymol edible coating had a significant effect on hyphal growth (*p* < 0.05) (Table 1) (Figure 4A), with a concentration of 95.10 mg/L for 50% maximal effect. According to the results of OD600 values, 40–2000 mg/L TKL had a significant inhibitory effect on the spore germination of *C. fioriniae* (Figure 4B). As the concentration increased, the inhibitory effect became more pronounced. When the concentration of TKL reached 1000 mg/L, the germination of its conidia could be completely inhibited. More importantly, the growth of *C. fioriniae* can be divided into four stages (Figure 4C), which were the lag phase (0–36 h), the logarithmic phase (36–96 h), the stable phase (96–120 h), and the decline phase (120–192 h). However, after thymol edible coating with a thymol concentration of 100 mg/L (TKL100) treatment, the growth of *C. fioriniae* slowed down rapidly, and the four growth stages of *C. fioriniae* became lag phase (0–36 h), logarithmic phase (36–96 h), and decline phase (96–192 h). After 48 h of culture, the growth rate of *C. fioriniae* was slowed, the dry weight of mycelium was only 80% of that of the control group (*p* < 0.05), and the decline phase was advanced by 24 h. Meanwhile, SEM observed that the normal *C. fioriniae* hyphae were relatively smooth and transparent and presented in the shape of strips: less curled and folded and not broken (Figure 4D). However, after treatment with TKL100, the hyphae of *C. fioriniae* showed obvious shrinkage; the surface was rough and curled and even had obvious fractures (Figure 4E).

### 3.4. Effects of TKL on C. fioriniae Enzyme Activities

Malondialdehyde (MDA) was the final decomposition product of membrane lipid peroxidation, and its content could reflect the degree of damage to microorganisms [52]. The content of MDA in *C. fioriniae* treated with TKL100 at 72 h, 96 h, and 120 h was significantly higher than that in the control group (*p* < 0.05), but there was no significant difference with the control group at 24 h and 48 h (*p* > 0.05) (Figure 5A). Because chitinase (CHI) can catalyze the hydrolysis of chitin, β-1,3-glucanase can catalyze the hydrolysis of β-1,3-glucan glycoside bonds [53]. As a consequence, when the activity of chitinase and β-1,3-glucosidase increases in cells, the polysaccharide component of the cell wall decreases, and the cell wall structure is damaged. In the present study, after TKL100 induction, the intracellular chitin hydrolase activity continued to increase, and with the prolongation of the induction time, the enzyme activity showed an upward trend that was initially fast and then slow. During the whole treatment period, the CHI activity of the TKL100 group was significantly higher than that of the control group (Figure 5B). At 24 h, the CHI activity of the TKL100 group was approximately 1.14 times that of the control group, reaching 158.74 U/L (*p* < 0.05). TKL100 was treated for 24 h, 48 h, 72 h, and 96 h; β-1,3-glucanase content was significantly increased (*p* < 0.05) (Figure 5C), and there was no significant difference between 120 h and the control group. Therefore, TKL100 might induce the production of CHI and β-1,3-glucosidase and promote the degradation of the cell wall of *C. fioriniae*. In addition, polygalacturonase (PG), cellulase (Cx), and pectin methylgalacturonase (PMG) are fungal metabolites that promote the degradation of their cell walls, leading to cell damage and apoptosis [6]. The PG activity of *C. fioriniae* was significantly increased after TK100 treatment throughout the culture cycle, especially at 48 h (Figure 5D), and the PG activity was increased by 43.6% compared with the control group (*p* < 0.05). After *C. fioriniae* was treated with TKL100 (Figure 5E), the Cx activity decreased significantly at 24 h; but at 48 h, 72 h, 96 h, and 120 h, the Cx activity gradually increased, which was significantly different from the control group (*p* < 0.05). In addition, after TKL100 treatment (Figure 5F), the 24 h PMG activity of *C. fioriniae* was not significantly different from that of the control group (*p* > 0.05) and then gradually increased, and the PMG activity at 120 h was 1.33 times that of the control group (*p* < 0.05). Therefore, TKL100 may promote the synthesis of PG, CX, and PMG and accelerate the metabolic process of the fungal cells, which promotes the early appearance of fungal senescence and also causes damage to the cell wall.

### 3.5. Effect of TKL on Surface Disease Changes in Okra Inoculated with the Pathogens

As shown in Figure 6A–D, the brown spot rate on the surface of okra gradually increased with the prolongation of storage time, and the wrinkle rate of the four treatment groups was significantly different from 1 to 6 days of storage (*p* < 0.05). The results indicate that CK-Y > TKL-Y > CK-W > TKL-W. After 10–14 days of storage, 100% of the okra in the four treatment groups had brown spots on the surface, and CK-Y had the brownest spots on the surface. At the same time, the wrinkle generation rate on the surface of okra also increased with the prolongation of storage time, and within 14 days of storage, there was a trend of CK-Y > TKL-Y > CK-W > TKL-W (*p* < 0.05), the okra folds of CK-Y are the most obvious. The rate of white hyphae on the surface of okra also gradually increased with the prolongation of storage time, and there were significant differences among the four treatment groups, especially between CK-Y and TKL-Y within 1–14 days. The silk production rate was significantly higher than that of TKL-Y (*p* < 0.05). In addition, after inoculation with pathogenic fungus, with the prolongation of storage time, the hardness of okra also changed significantly, and the hardness of the four treatment groups all showed a downward trend; CK-Y in particular decreased the most sharply, and TKL-W decreased the most (*p* < 0.05).

### 3.6. Effect of TKL on Lignin Synthesis Pathway in Okra Inoculated with the Pathogens

#### 3.6.1. Total Phenols and Total Flavonoids

As shown in Figure 7A, B, on the first day, the total phenolic content of okra in the four treatment groups presented as TKL-W > CK-W > TKL-Y > CK-Y (*p* < 0.05); on the second day, the total phenolic content of TKL-Y and CK-Y decreased significantly. On the third day, the contents of TKL-Y and CK-Y increased, showing the order of TKL-W > TKL-Y > CK-Y > CK-W (*p* < 0.05). On the fifth day, the total phenolic content of the TKL-Y group decreased, whereas that of the CK-Y group increased, showing TKL-W, TKL-Y, and CK-W > CK-Y. On the sixth and seventh days, CK-Y continued to rise, showing CK-W > TKL-W > CK-Y > TKL-Y (*p* < 0.05). On the eighth day, the total phenolic content of both TKL-Y and CK-Y reached the maximum; on the ninth day, the total phenols and total flavonoids of TKL-Y and CK-Y sharply decreased. However, on the tenth day, it was TKL-Y > TKL-W > CK-W > CK-Y (*p* < 0.05). In a word, the total phenols and flavonoid contents of okra treated by TKL coating were higher than those of the control group.

#### 3.6.2. Key Enzyme Activities in the Lignin Synthesis Pathway

As shown in Figure 8A, on the first, second, fourth, and eighth days, the PAL enzyme activities of the four treatment groups were all presented as TKL-Y > TKL-W > CK-Y > CK-W (*p* < 0.05). The trend of PAL enzyme activity was the highest in the TKL-Y group. However, on the tenth day, it was TKL-Y > TKL-W and CK-Y > CK-W (*p* < 0.05). As shown in Figure 8B, for the first, second, third, fourth, and fifth days, the 4CL enzyme activity of the four treatment groups presented as TKL-Y > TKL-W > CK-W > CK-Y *(p* < 0.05). The TKL-Y group had the highest enzyme activity. However, on the seventh, eighth, ninth, and tenth days, TKL-W > CK-W > CK-Y > TKL-W (*p* < 0.05), and TKL-W enzyme activity was the highest. As shown in Figure 8C, on second, third, and fourth days, the C4H of the four treatment groups showed a trend of TKL-Y > CK-Y > TKL-W > CK-W (*p* < 0.05). The TKL-Y group had the highest enzyme activity. The results indicated that TKL-Y > CK-W > CK-Y > TKL-W (*p* < 0.05) on the tenth day. The PAL, 4CL, and C4H enzyme activities showed an increasing trend after the TKL100 coating.

#### 3.6.3. Okra Lignin Content

As shown in Figure 8D, on the 2nd day, the lignin content of the four treatment groups presented as CK-W > CK-Y > TKL-W and TKL-Y (*p* < 0.05), and on the 3rd day the lignin content of the four treatments groups There was no significant difference in content (*p* > 0.05). However, on the 4th day, TKL-W had the highest lignin content, CK-Y content is the lowest (*p* < 0.05), and on the 5th day CK-Y > TKL-W > TKL-Y > CK-W (*p* < 0.05). After being placed at 28 °C for 6 days, the lignin content of TKL-W reached the highest 65.62 ± 0.68 mg/g, much higher than other groups and other storage days (*p* < 0.05). At 6–10 days, no obvious rule was found in the four groups, but the overall trend was as follows: After treating okra with TKL coating, the lignin content of okra showed an upward trend.

## 4. Discussion

Easy to grow, unique in taste and rich in nutrients, okra is a very popular vegetable with high market value [54]. However, postharvest fungal diseases limit the storability and shelf life of okra and cause serious economic losses during postharvest storage and transportation. In particular, the occurrence of lignification during postharvest storage seriously affects the okra edible value. In addition, okra is susceptible to postharvest attacks by fungus, such as *Fusarium*, *Alternaria alternate*, and *Aspergillus niger* [7,55]. We isolated an anthracnose pathogen from postharvest okra that was naturally decaying, identified it as *C. fioriniae*, and confirmed the pathogenicity of *C. fioriniae* to okra. In addition, when studying the biological characteristics of *C. fioriniae*, it was found that mannitol, sucrose, and starch were the optimal carbon sources. Potassium nitrate was the optimal nitrogen source. When pH = 7, the growth rate was the fastest; when the NaCl concentration was 0.5%, the growth rate was the fastest. The best growth was at 28 °C; light is favorable for the growth of *C. fioriniae*. These coincide with the characteristics of *Colletotrichum acutatum*, the bacterium responsible for Belgian apple bitter rot [56]. Clarifying the biological characteristics of *C. fioriniae* could provide technical support and a theoretical basis for the prevention and control of anthrax disease in okra [57].

In recent years, a lot of research and development of plant extracts have been carried out as substitutes for synthetic fungicides for the management of postharvest decay in fruits and vegetables [58]. Thymol has been reported to be very effective in controlling various postharvest diseases of fruits and vegetables [59]. Thymol showed good antibacterial activity in the postharvest disease control of fruits and vegetables and had a strong antibacterial effect [60]. For example, citral and carvacrol inhibited the growth of Botrytis cinerea at a concentration of 250 mg/L [61], whereas thymol also inhibited the growth of Botrytis cinerea at a concentration of 150 mg/L. At 1500 and 500 mg/L, thymol could completely inhibit the hyphal growth of 17 phytopathogenic fungi, and its antibacterial effect was stronger than that of benomyl [62]. Meanwhile, according to previous studies, thymol can cause structural damage to the cell wall and membrane of fungi, as it has shown strong antibacterial and antifungal properties [63,64]. For instance, the inhibitory mechanism of thymol on *E. sakazaki* is that it leads to the depolarization of the cell membrane, the decrease of intracellular ATP concentrations, and the decrease of pH, which indicates that thymol mainly causes the cell membrane rupture of *E. sakazaki* and the destruction of its homeostasis [65]. Moreover, thymol showed extensive antifungal activity against *Aspergillus niger*, *Neopestalotiopsis sp.*, *Alternaria alternata*, *Penicillium sp.*, *Cladosporium xylophilum*, and *Botrytis cinerea* isolates on blueberry. Furthermore, thymol eventually causes cell membranes to break down and cell material to leak out of these postharvest blueberry pathogens, eventually leading to cell death [66]. Nano-thymol emulsion showed significant antifungal activity against *B. cinerea*, the main pathogen of postharlife gray mold of tomato. ROS of *B. cinerea* can be induced to accumulate in the mycelium, further leading to lipid peroxidation, cell membrane damage, and subsequent cell death [67]. Therefore, thymol mainly caused damage to the cell membrane, cell wall, and metabolism of postharvest pathogens in fruits and vegetables, which inhibits growth, reproduction, and even death.

Our results show that TKL has a good inhibitory effect on the okra anthracnose pathogen *C. fioriniae*, which can significantly inhibit the growth and reproduction of its hyphae. At the same time, after treatment with TKL100, the stationary phase of *C. fioriniae* was shortened, the decline phase was advanced, and the growth limit was reduced. In addition, SEM observation showed that, compared with the control group, the hyphae of *C. fioriniae* shrank, and the surface was rough and even broken after TKL100 treatment. This indicates that TKL can disrupt the structure of *C. fioriniae* and inhibit hyphal growth and spore germination.

Studies have shown that the higher the MDA content, the higher the degree of peroxidation of the fungal cell membrane, and the more serious the damage will be [68]. After TKL treatment, the MDA content was significantly higher than that in the control group at 72 h, 96 h, and 120 h, but there was no significant difference at 24 and 48 h. This indicated that TKL started to induce the peroxidation of the *C. fioriniae* cell membrane after 48 h, causing certain damage to the cell membrane and, thus, achieving the bacteriostatic effect. Chitinase (CHI) and β-1,3-glucanase are enzymes closely related to the formation of the cell wall of *C. fioriniae* [69,70]. Chitinase can catalyze the hydrolysis of chitin and β-1,3-glucan. When the activities of chitinase and β-1,3-glucosidase increased in *C. fioriniae* cells, the polysaccharide composition of the cell wall was reduced, and the cell wall structure was damaged [71]. After TKL100 induction, the chitin hydrolase activity in *C. fioriniae* cells continued to increase, and with the prolongation of induction time, the enzyme activity showed a trend of increasing rapidly initially and then slowly. The CHI activity of the TKL100 group was significantly higher than that of the control group throughout the treatment period. Especially at 24 h, the CHI activity of the TKL100 group was about 1.14 times that of the control group, reaching 158.74 U/L. In addition, the content of β-1,3-glucanase was significantly higher than that in the control group when TKL100 was treated for 24 h, 48 h, 72 h, and 96 h, respectively. Therefore, TKL100 promotes the degradation of the *C. fioriniae* cell wall by inducing the production of CHI and β-1,3-glucosidase in the *C. fioriniae* cells, resulting in cell death and rupture. All these coincide with the antifungal mechanism of thymol [72].

In addition, PG, Cx, and PMG, which are fungal metabolites, can also promote cell wall degradation, leading to cell damage and apoptosis [73,74,75,76]. After treatment with TK100, the PG activity of *C. fioriniae* was significantly increased, especially at 48 h by 43.6%. At 24 h, the Cx activity decreased significantly, but at 48 h, 72 h, 96 h, and 120 h, the Cx activity gradually increased, which was significantly different from the control group. In addition, after TKL100 treatment, the PMG activity was not significantly different from the control group at 24 h, and then gradually increased. The PMG activity at 120 h was 1.33 times that of the control group. Therefore, TKL100 may accelerate the metabolic process of *C. fioriniae*; promote the synthesis of PG, CX, and PMG; promote the early appearance of the recession period; and also cause irreversible damage to the cell wall. These results indicate that *C. fioriniae* may degrade the cell wall of okra plant cells by secreting polygalacturonase, cellulase, and pectin methylgalacturonase during the process of infecting okra [77].

All of the above confirmed that TKL has a good inhibitory effect on the okra anthracnose pathogen *C. fioriniae*, and its antifungal mechanism may be mainly to inhibit fungal growth, metabolic process, and spore germination [72,78]. Second, it induces cell membrane peroxidation and promotes cell wall degradation, thereby promoting the apoptosis of *C. fioriniae*. In exploring the prevention and control mechanism of TKL on okra anthracnose disease, we found that *C. fioriniae* could cause anthracnose disease, folds, brown spots, and depressions to both injured and intact okra at 28 °C. However, after TKL100 coating treatment, it was found from the surface observation that the symptoms of anthracnose disease, the rate of brown spots, the rate of wrinkles, and the production of white hyphae on the surface of okra were significantly lower than those of the control group. In the meantime, testing of the hardness showed that the hardness of wounded and intact okra after TKL100 coating was significantly higher than that of the control group within 1–14 d at 28 °C. From the surface disease observation and hardness test, it can be shown that TKL100 can not only inhibit infection of an intact okra by *C. fioriniae* to a certain extent but also inhibit the appearance of okra tissue softening and depression; therefore, TKL100 has a certain ability to prevent and control okra anthracnose disease.

To explore whether the potential mechanism of TKL100 controlling okra anthracnose disease is related to the metabolism of okra lignin, we started from the synthesis pathway of lignin, starting with phenols, and found that the total phenolic content of okra treated with TKL100 was higher than that of the control group, especially when placed at 28 °C for 10 days. The total phenolic content of intact okra treated with TKL100 was the highest, and the total phenolic content of untreated injured okra was the lowest. Secondly, the changes in total flavonoids also showed a similar trend. When placed at 28 °C for 10 days, the total flavonoid content of the intact okra treated with TKL100 was the highest, but the total flavonoid content of the untreated intact okra was the lowest. It was consistent with the regulatory trend of lignin biosynthesis pathways reported by Xie et al. [79]. This indicates that TKL100 is beneficial to the accumulation of total phenols in intact and injured okra and provides more preconditions for lignin synthesis. Meanwhile, it can also promote the synthesis of total flavonoids to a certain extent, but the effect is weaker than that of total phenols. With more lignin precursors, we further explored the effect of TKL100 on key enzyme activities (PAL, C4H, and 4CL) in the lignin synthesis process. The PAL, C4H, and 4CL activities of okra were significantly higher than those of untreated okra, but there was no significant difference in enzyme activities between wounded and intact okra, which may be due to the relative effect of wounded treatment on improving the enzyme activities of PAL, C4H, and 4CL. The processing of TKL100 is too weak, so there is no obvious trend of unity. However, it can still be shown that the treatment of okra with TKL100 coating can stimulate the improvement of the key enzyme activities of okra-related lignin synthesis and promote the synthesis of lignin. It is consistent with the key enzyme activity of promoting lignin accumulation in loquat fruit and ramie [80,81].

Therefore, to verify whether TKL100 coating promotes the synthesis of okra lignin, we measured the lignin content of okra and placed it at 28 °C for 1–10 days. The content of lignin was higher than that of the untreated group, especially on the sixth day. The lignin content of the intact okra treated with TKL100 coating showed a peak state, which was significantly higher than that of the other groups by 2.24 times. However, on the 10th day, the lignin content of the TKL100 treated okra after the coating treatment was lower than that of the untreated group, which may be the reason why the okra began to rot in the late storage period, and the lignin of all okra began to show a downward trend. It may be that the okra begins to lose defenses against microbial infestation.

In conclusion, we can conclude that TKL has a good inhibitory effect on the growth and reproduction of the okra anthracnose pathogen *C. fioriniae*; its inhibitory effect is achieved mainly by destroying the cell membrane of *C. fioriniae*, degrading the cell wall, and affecting the metabolism. In addition, TKL100 coating of okra can prevent the anthracnose pathogen *C. fioriniae* from infecting okra and delay the spread of disease symptoms to a certain extent; its prevention and control mechanism is related to the regulation of the okra lignin synthesis pathway to a certain extent. It may be by promoting total synthesis of phenol and total flavonoids and stimulation of enzyme activity to promote lignin synthesis that it achieves better stress resistance and disease control effects. However, this may only be one of the potential prevention and control mechanisms, and more molecular metabolic pathways and gene targets need to be explored to verify this trend. In the later stage, it will be further verified that TKL can control okra anthracnose by regulating the lignin synthesis pathway.

## 5. Conclusions

An anthracnose pathogen was isolated from okra and identified as *C. fioriniae*; *C. fioriniae* has strong pathogenicity to okra and is the main pathogen causing okra anthracnose. The results showed that TKL inhibited the growth of *C. fioriniae*, and TKL100 inhibited the growth and reproduction of *C. fioriniae* by causing damage to the cell membrane, cell wall, and cell metabolism in different degrees. In addition, TKL100 coating treatment could significantly reduce the penetration of *C. fioriniae* to okra and reduce the rate of brown spot, fold, and mycelium growth on the okra surface. The synthesis of total phenols and total flavonoids in the TKL group increased, and the activities of PAL, C4H, and 4CL enzymes related to lignin synthesis increased. The lignin content of TKL-W increased significantly, especially after 6 days in a 28 °C incubator. The lignin content of TKL-W was the highest, reaching 65.62 ± 0.68 mg/g, which was 2.24 times that of CK-W. Therefore, it can be concluded that TKL inhibits the growth and reproduction of *C. fioriniae* mainly by destroying the cell membrane of *C. fioriniae*, degrading the cell wall, and affecting metabolism. TKL100 coating okra can regulate the okra lignin synthesis pathway to a certain extent to prevent anthrax. The pathogen *C. fioriniae* infects okra and delays the spread of disease symptoms, achieving better stress resistance and disease control effects. However, this may only be one of the potential prevention and control mechanisms, and more molecular metabolic pathways and gene targets need to be explored to verify this trend. In a later stage, it will be further verified that TKL can control okra anthracnose by regulating the lignin synthesis pathway.

## 6. Patents

The research involved a patent for an invention: “A thymol compound biological coating preservative and its preparation method and application” (CN2010919222.9).

## Figures and Tables

**Figure 1 foods-12-00395-f001:**
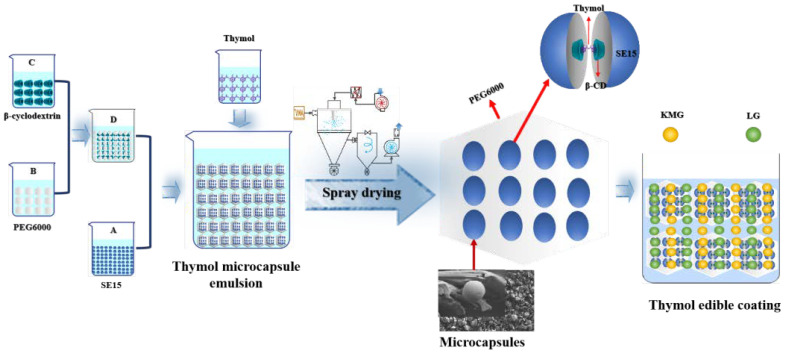
Flow chart of TKL coating solution preparation. β-cyclodextrin (β-CD); polyethylene glycol 6000 (PEG6000); and sucrose fatty acid ester (SE15).

**Figure 2 foods-12-00395-f002:**
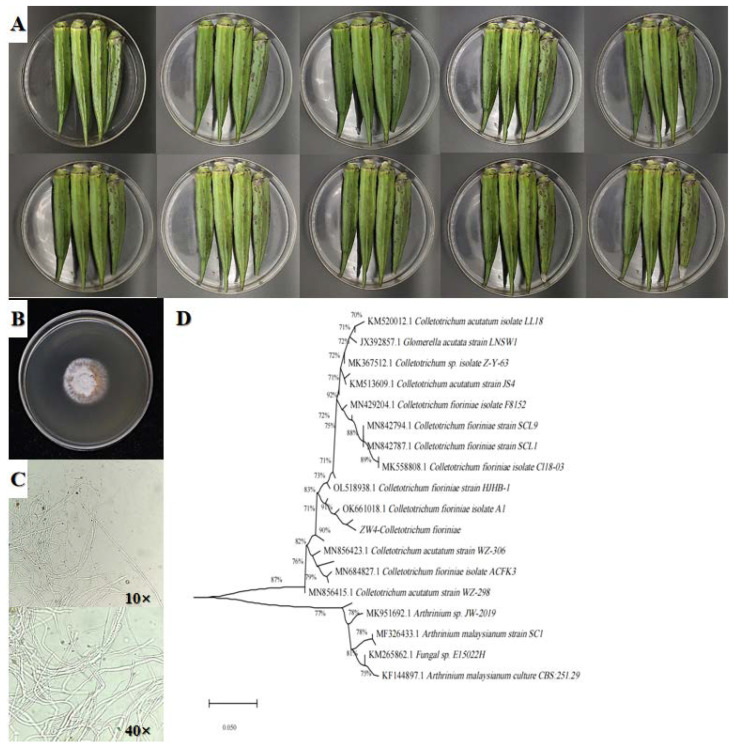
Symptoms of anthracnose in okra stored at 28 °C for 1–10 days (**A**), morphological characteristics of isolated pathogens (**B**,**C**), and phylogenetic tree of the pathogens (ZW4-*C. fioriniae*) responsible for okra anthracnose (**D**).

**Figure 3 foods-12-00395-f003:**
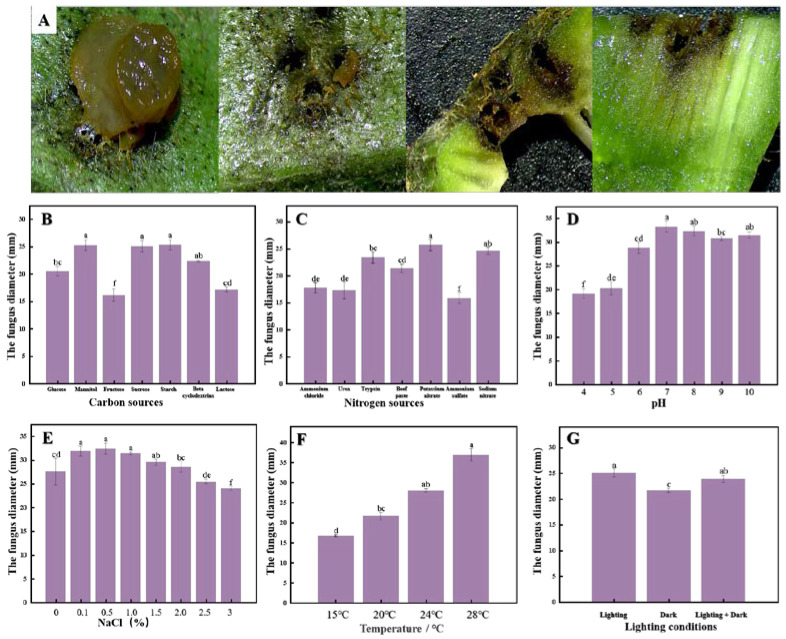
Okra disease under the microscope after being placed at 28 °C for 3 days (**A**), colony diameter under 7 carbon sources (**B**), colony diameter under 7 nitrogen sources (**C**), colony diameter under different pH conditions (**D**), colony diameter under different NaCl concentrations (**E**), colony diameter at different temperatures (**F**), and colony diameter under different light conditions (**G**). SPSS was used for significance analysis and letter marking of significant difference was used.

**Figure 4 foods-12-00395-f004:**
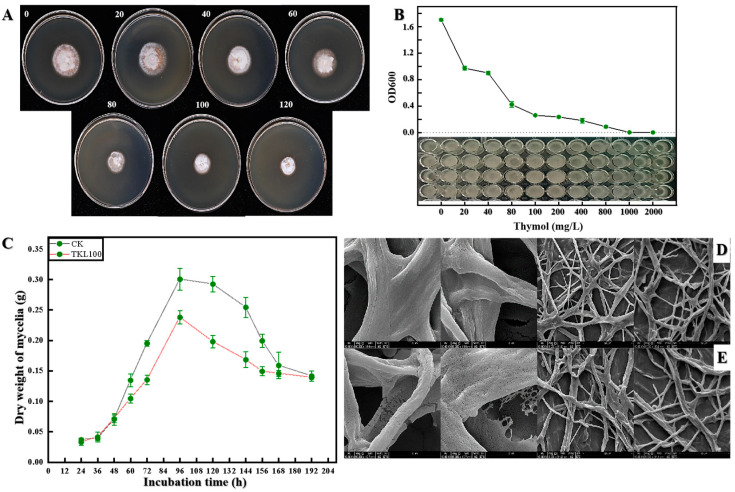
Colony growth after TKL treatment with different thymol concentrations, such as TKL 100 as a thymol edible coating with a thymol concentration of 100 mg/L. (**A**); determination of minimum inhibitory concentration (MIC) of TKL on the growth of pathogenic fungus (**B**); changes in the growth cycle of *C. fioriniae* after TKL100 treatment (**C**); healthy hyphae of *C. fioriniae* (**D**); and hyphae of *C. fioriniae* treated with TKL100 (**E**).

**Figure 5 foods-12-00395-f005:**
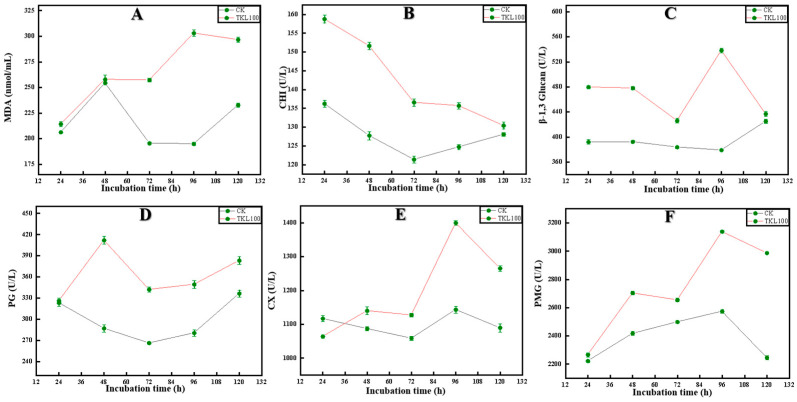
After TKL100 treatment for 24 h, 48 h, 72 h, 96 h, and 120 h, compared with the control group, MDA changes of *C. fioriniae* (**A**), CHI changes after TKL100 treatment (**B**), β-1,3-glucanase changes (**C**), PG changes (**D**), CX changes (**E**), and PMG changes (**F**).

**Figure 6 foods-12-00395-f006:**
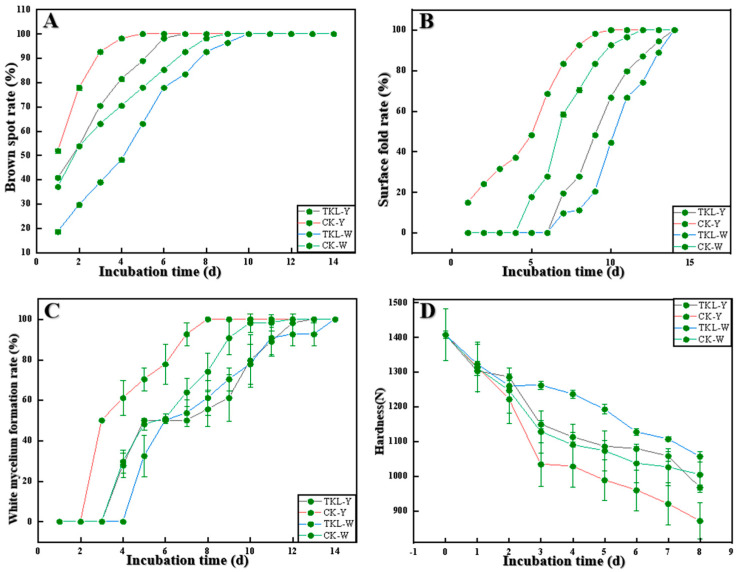
The rate of brown spots of okra in four groups was changed at 28 °C for 0–14 days (**A**), the rate of the surface fold of okra in four groups (**B**), the rate of white hyphae formation of okra in four groups (**C**), and the hardness changes of okra in four groups (**D**).

**Figure 7 foods-12-00395-f007:**
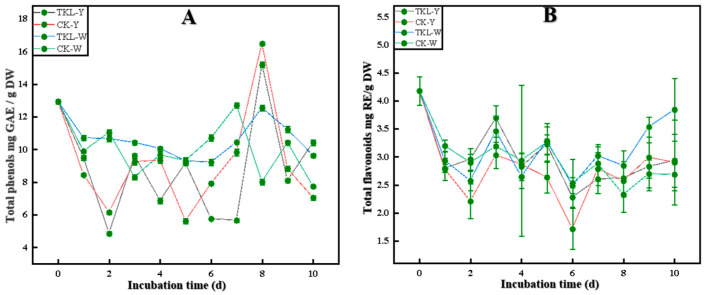
Total phenol content changes of okra in four groups at 28 ℃ for 0-10 days (**A**); total flavonoid content changes of okra in four groups (**B**).

**Figure 8 foods-12-00395-f008:**
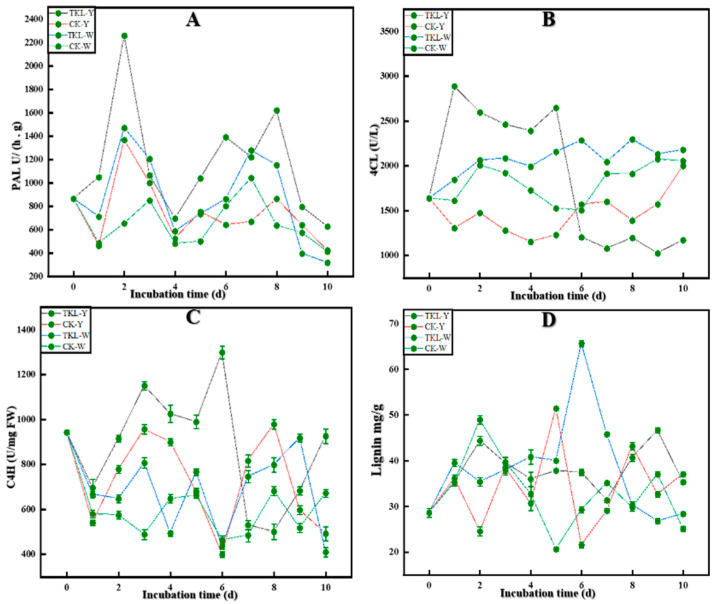
PAL changes of okra in four groups at 28 °C for 0–10 days (**A**), 4CL changes of okra in four groups (**B**), C4H changes of okra in four groups (**C**), and okra lignin content changes in four groups (**D**).

**Table 1 foods-12-00395-t001:** Inhibitory effect of TKL at different thymol concentrations on *C. fioriniae*.

Bacteriostatic Agent	Thymol Concentration	the Fungus Diameter	Inhibition Rate
TKL 0	0 mg/L	30.12 ± 0.59 ^a^	0.00%
TKL 20	20 mg/L	26.67 ± 0.77 ^ab^	11.45%
TKL 40	40 mg/L	24.23 ± 1.71 ^bc^	19.57%
TKL 60	60 mg/L	18.00 ± 1.00 ^cd^	40.25%
TKL 80	80 mg/L	17.12 ± 1.46 ^de^	43.16%
TKL 100	100 mg/L	14.97 ± 0.73 ^ef^	50.32%
TKL 120	120 mg/L	11.81 ± 0.57 ^g^	60.81%

SPSS was used for significance analysis and letter marking of significant difference was used.

## Data Availability

The data presented in this study are available on request from the corresponding author. The data are not publicly available due to the ongoing nature of the experiment.

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
