# Peer review of "Thymol Edible Coating Controls Postharvest Anthracnose by Regulating the Synthesis Pathway of Okra Lignin"

_foods, 2023, doi:10.3390/foods12020395_

Round 1

Reviewer 1 Report

Comments

             Zhang and co-workers' research findings on Thymol/KGM/LG edible coating solution showed a growth inhibition effect on Colletotrichum fioriniae and controlled postharvest anthracnose by regulating the synthesis pathway of okra lignin. The authors provided some good results about the inhibition of Colletotrichum fioriniae. However, the presentation of the manuscript is not up to the standard of the Journal. The authors should improve the introduction and discussion sections of this manuscript. In addition, I have the following comments and suggestions to improve the quality of their manuscript:

1.        The abstract section should contain objective and novelty of the study, specific methods and important findings with conclusion. Rewrite the abstract section.

2.        In the abstract, line 14, double check the result “EC50 was 95.10 mg/L, MIC was 1000 mg/L”.

3.        Why did you specifically choose thymol in this study? So add more details about the importance of thymol in the introduction section. Clearly explain the importance of this study.

4.        Further, provide more information about anthracnose disease and the pathogen Colletotrichum fioriniae in the introduction section.

5.        In Materials and Methods, sub section 2.2, authors mentioned that the pathogenic bacteria were isolated and purified? Bacteria or Fungi? Lines 133, 136, 137, 144, 158, 159, 162, 167, 171, 172, 205, 288, 292, 299, 311, 318, 372, 414, 415, 429, 506, 507, 555. These kinds of critical errors should be corrected.

6.        Did authors evaluate the antifungal activity of thymol itself without formulation? What is the MIC of thymol against Colletotrichum fioriniae? If not evaluate the MIC of thymol. Then only readers can easily understand the activity of Thymol/KGM/LG edible coating.

7.        Also separate treatment using thymol should be included for postharvest treatment of okra.

8.        Place Figure 4F – K under the section 3.4, similarly, place Figure 5E and F under the section 3.6.1.

9.        Correct the title of the sub section 3.6.2 “Total phenols and total flavonoids” for enzyme activity.

10.    The discussion section should be improved by adding previous studies in relation to mechanisms of action of antifungal activities of Thymol and other related compounds. Add more citations to confirm the findings of the study.

11.    After first mentioning Colletotrichum fioriniae in the Introduction section, Use abbreviation of Colletotrichum fioriniae (C. fioriniae) throughout the manuscript.

12.    Provide high resolution figure of Phylogenetic Tree (Figure 2D) and Graphs (Figure 3).

13.    Conclusion section should be reduced. Only highlight the important findings of the work with future perspectives.

14.    Italicize all the scientific names throughout the manuscripts (Example: Lines 39, 305, 306, 313, 320, 323, 328, 333, 382, 390, 401, 404, 407, 411, 752 etc.). In particular, scientific names are not italicized in the References section.

Reviewer 2 Report

The novelty of the manuscript is accepted however, a few comments are recommended as follows to improve the quality of the paper.

Please reconsider the title. Put the full name of abbreviation in the title.

Abstract: Line 8-12: ‘The growth of Colletotrichum fioriniae was observed to be divided into four cycles; mannitol, sucrose and starch were the optimum carbon sources; potassium nitrate was the optimum nitrogen source; when pH = 7, the growth rate was the fastest; NaCl concentration. When it 10 was 0.5%, fastest growing; the fastest growth occurs at 28 °C; the light was favorable for the growth of Colletotrichum fioriniae’.

Rewrite this section. It is not clear.

Write the full name of Thymol/KGM LG and then mention abbreviation.

In the introduction, please describe the importance of edible coating in postharvest of fruits and vegetables.

Materials and Methods: Describe more details regarding the thymol microcapsule emulsion. Address Figure 1 in the text.

For Determination of total phenols and total flavonoids: please describe how fruit extract was prepared?

For results and discussion section, it should be mentioned that this part needs to reconsider carefully. The discussion section of results is poor. For example for the effect of different NaCl (%) on colony growth (Line 328) the results were reported. It is crucial to add discussion.

Also it is important and necessary to compare your findings with other available information.

Round 2

Reviewer 1 Report

The authors revised the manuscript according to the reviewers’ comments. In my opinion, this manuscript is suitable for publication. 

In Lines 49 and 58, italicize the scientific names.

In the References section, the authors did not italicize the scientific names. Correct it before the publication of the manuscript.

Author Response

Revised, Thank you very much.

Reviewer 2 Report

The corrections were made.

Author Response

Thank you very much for taking the time to review our manuscript.